# Derivative Estimation in Random Design

**Yu Liu[1], Kris De Brabanter[1,2*]**

[1]Department of Computer Science, [2]Department of Statistics

## Abstract

We propose a nonparametric derivative estimation method for random design
without having to estimate the regression function. The method is based on a
variance-reducing linear combination of symmetric difference quotients. First, we
discuss the special case of uniform random design and establish the estimator's
asymptotic properties. Secondly, we generalize these results for any distribution of
the dependent variable and compare the proposed estimator with popular estima-
tors for derivative estimation such as local polynomial regression and smoothing
splines.

## 1 Introduction

In the area of statistics, nonparametric regression is often of great interest due to its flexibility
and different regression methods have been fully explored [1, 2, 3]. Derivative estimation has
received less attention than regression and it is often treated as the "by-product" of nonparametric
regression problems e.g. local polynomial regression [1] and smoothing splines [4]. Derivatives are
widely used in different areas, for example, analyzing human growth data [5, 6]. Other applications
include exploring the structure of curves [7, 8], analyzing significant trends [9], comparing regression
curves [10], characterization of nanoparticles [11], neural network pruning [12], estimating the leading
bias term in the construction of confidence intervals [13, 14] and bandwidth selection methods [15].

In general, derivative estimators can be divided into three categories: local polynomial regression,
regression/smoothing splines and difference quotients [16]. Local polynomial regression relies on the
Taylor expansion and the coefficients, obtained by solving a weighted least squares problem, provide
estimates of the derivatives. Asymptotic properties for the regression as well as the derivatives
are given in [1]. Derivative estimation via smoothing splines is obtained by differentiating the
spline basis [17]. These estimators are shown to achieve the optimal $L_2$ convergence rate [18] and
asymptotic properties are discussed [19]. For the latter, the smoothing parameter selection is quite
difficult. Especially for smoothing splines whose parameter depends on the order of the derivative [4].
Difference quotient based derivative estimators [16, 20] provide a noisy version of the derivative
which could be smoothed by any nonparametric regression method. The difference estimator proposed
by [16] is quasi unbiased but the variance is $O(n^2)$ where $n$ is the sample size. In order to reduce
the variance, [21, 22] proposed a variance-reducing linear combination of $k$ symmetric difference
quotients, where $k$ is considered to be a tuning parameter.

More recently, [23] proposed a sequence of approximate linear regression representations in which
the derivative is just the intercept term. Although their results are very appealing, they rely on rather
stringent assumptions on the regression function. These assumptions are relaxed in [24] where a linear
combination of the dependent variables are used to obtain the derivatives. The variance reducing
weights are obtained by solving a constraint optimization problem for which a closed form solution
is derived. Also, they showed that the symmetric form used in [21] and [22] reduces the order of

`yuliu@iastate.edu, kbrabant@iastate.edu`

estimation variance without significantly increasing the estimation bias in the interior. They proposed an asymmetric estimator for the derivatives at the boundaries. All results from [23] and [24] assume the equispaced design and both authors do not mention the extension to the random design setting. In this paper we extend the difference quotient based estimator to the random design and provide a framework that can be used to extend other difference based estimators to the random design. Further, we show that the extension from equispaced to random design for higher order derivatives is not trivial. Simply using the estimator from [21] and [22] in random design will lead to an inconsistent estimator. In the simulation study, we show that the new estimator has similar performance compared to local polynomial regression and penalized smoothing splines. All the proofs of the lemmas and theorems can be found in Supplementary Material accompanying the paper.

## 1.1 Equispaced design vs. random design

Consider the data $(X_1, Y_1),\ldots,(X_n, Y_n)$ which form an independent and identically distributed (i.i.d.) sample from a population $(X, Y)$, where $X_i \in \mathcal{X} = [a, b] \subseteq \mathbb{R}$ and $Y_i \in \mathbb{R}$ for all $i = 1, \ldots, n$. In the equispaced design case, the response variables are assumed to satisfy

$$Y_i = m(x_i) + e_i, \quad i = 1, \ldots, n, \tag{1}$$

where $x_1, \ldots, x_n$ are nonrandom and $x_{i+1} - x_i = (b-a)/(n-1)$ is constant for all $i$. In this setting, the regression function is given by $m(x) = \mathbf{E}[Y]$ and we assume that $\mathbf{E}[e] = 0$ and $\mathbf{Var}[e] = \sigma_e^2 < \infty$. In contrast to the equispaced design, the design points $X$ in random design are random variables and are generated from an unknown distribution $F$. Consider the following model

$$Y_i = m(X_i) + e_i, \quad i = 1, \ldots, n, \tag{2}$$

where the regression function is given by $m(x) = \mathbf{E}[Y|X = x]$ and assume that $\mathbf{E}[e] = 0$, $\mathbf{Var}[e] = \sigma_e^2 < \infty$, $X$ and $e$ are independent. The derivative estimators discussed in [21, 22, 23, 24] use the symmetric property $x_{i+j} - x_i = x_i - x_{i-j}$ since they assume equispaced design. However, in the random design this property no longer holds and it also presents extra theoretical difficulties as we will show in the next sections.

## 2 Difference based derivative estimators based on order statistics

The first difference quotients were proposed by [16] for fixed design. Extending their estimator to random design yields

$$\hat{q}_i^{(1)} = \frac{Y_i - Y_{i-1}}{X_i - X_{i-1}} \tag{3}$$

which is a noise corrupted version of the first order derivative in $X_i$. Although this estimator is quasi unbiased, two problems immediately emerge: (i) no simple expression for the difference $X_i - X_{i-1}$ is available to study its asymptotic properties; (ii) the variance is proportional to $n^2$ (see next section). In order to discuss the asymptotic properties of this different quotient, we need to obtain an asymptotic expression for the difference $X_i - X_{i-1}$ which is not trivial in the random design setting. However, in a special case i.e., $X = U \sim \mathcal{U}(0, 1)$ and arranging the random variables in order of magnitude according to $U$ (order statistics), the asymptotic properties of the difference can be obtained using order statistics (see Lemma 1). In what follows, $\mathcal{U}(0, 1)$ denotes the uniform distribution between 0 and 1.

## 2.1 Approach based on order statistics

Consider $n$ bivariate data forming an i.i.d sample from a population $(U, Y)$ and further assume $U \sim \mathcal{U}(0, 1)$. Arrange the bivariate data $(U, Y)$ in order of magnitude according to $U$ i.e., $U_{(1)} < U_{(2)} < \ldots < U_{(n)}$ where $U_{(i)}$, $i = 1, \ldots, n$ is the $i$th order statistic. Consider the following model:

$$Y_i = r(U_{(i)}) + e_i, \tag{4}$$

where $r(u) = \mathbf{E}[Y|U = u]$ is the regression function and assume $\mathbf{E}[e] = 0$, $\mathbf{Var}[e] = \sigma_e^2 < \infty$, $U$ and $e$ are independent. Our goal is to obtain a smoothed version of the first order derivative of $r$.

Since the estimator (3) suffers from high variance, [21] and [22] proposed a variance reducing linear combination of symmetric difference quotients. Our proposed extension to the random design

involving uniform order statistics is

$$\hat{Y}_i^{(1)} = \sum_{j=1}^{k} w_{i,j} \cdot \left( \frac{Y_{i+j} - Y_{i-j}}{U_{(i+j)} - U_{(i-j)}} \right), \tag{5}$$

where the weights $w_{i,1}, \ldots, w_{i,k}$ sum up to one. To avoid division by zero we require no ties, i.e. $U_{(l)} \neq U_{(m)}$ for $l \neq m$. Note that (5) is valid for $k + 1 \leq i \leq n - k$ and hence $k \leq (n-1)/2$. For the boundary regions i.e., for $2 \leq i \leq k$ and $n - k + 1 \leq i \leq n - 1$, the estimator (5) needs to be modified and is discussed in Section 2.5. A minor point is that the estimator (5) does not provide results for $\hat{Y}_1^{(1)}$ and $\hat{Y}_n^{(1)}$. One can ignore these two points from consideration. Proposition 1 shows the optimal weights $w_{i,j}$ which minimize the variance of (3).

**Proposition 1** *For $k + 1 \leq i \leq n - k$ and under model* (4)*, the weights $w_{i,j}$ that minimize the variance of* (5)*, satisfying $\sum_{j=1}^{k} w_{i,j} = 1$, are given by*

$$w_{i,j} = \frac{(U_{(i+j)} - U_{(i-j)})^2}{\sum_{l=1}^{k}(U_{(i+l)} - U_{(i-l)})^2}, \quad j = 1, \ldots, k. \tag{6}$$

At first sight, these weights seem to be different than the weights obtained by [21] and [22] for the equispaced design case. However, for the equispaced design case, plugging in the difference $u_{i+j} - u_{i-j} = 2j(b-a)/(n-1)$ in the weights obtained in Proposition 1 gives

$$w_{i,j} = \frac{(u_{i+j} - u_{i-j})^2}{\sum_{l=1}^{k}(u_{i+l} - u_{i-l})^2} = \frac{\frac{4j^2}{(n-1)^2}}{\frac{4}{(n-1)^2}\sum_{l=1}^{k}l^2} = \frac{6j^2}{k(k+1)(2k+1)}$$

which are exactly the weights used in the equispaced design. This shows that the weights for equispaced design are a special case of the weights in Proposition 1. To reduce the variance, for a fixed $i$, the $j^{th}$ weight (6) is proportional to the inverse variance of $\frac{Y_{i+j}-Y_{i-j}}{U_{(i+j)}-U_{(i-j)}}$ in (5).

Next, we need to find an asymptotic expression for the differences $U_{(i+l)} - U_{(i-l)}$. From [25, p. 14], the distribution of the difference of uniform order statistics is:

$$U_{(s)} - U_{(r)} \sim \text{Beta}(s - r, n - s + r + 1) \quad \text{for } s > r.$$

This result immediately leads to the lemma below.

**Lemma 1** *Let $U \overset{i.i.d.}{\sim} \mathcal{U}(0,1)$ and arrange the random variables in order of magnitude $U_{(1)} < U_{(2)} < \cdots < U_{(n)}$. Then, for $i > j$*

$$U_{(i+j)} - U_{(i-j)} = \frac{2j}{n+1} + O_p\left(\sqrt{\frac{j}{n^2}}\right), \qquad U_{(i+j)} - U_{(i)} = \frac{j}{n+1} + O_p\left(\sqrt{\frac{j}{n^2}}\right)$$

*and*

$$U_{(i)} - U_{(i-j)} = \frac{j}{n+1} + O_p\left(\sqrt{\frac{j}{n^2}}\right).$$

We briefly show why (3) suffers from high variance. Assume $r(\cdot)$ is twice continuously differentiable on $[0, 1]$, a Taylor expansion of $r(U_{(i \pm j)})$ in a neighborhood of $U_{(i)}$ gives

$$r(U_{(i \pm j)}) = r(U_{(i)}) + r^{(1)}(U_{(i)})(U_{(i \pm j)} - U_{(i)}) + O_p\left(\frac{j^2}{n^2}\right). \tag{7}$$

Applying Lemma 1 and (7) to (3), then for $n \to \infty$ we have

$$\mathbf{E}[\hat{q}_i^{(1)} | U_{(i-1)}, U_{(i)}] = \mathbf{E}\left[ \frac{Y_i - Y_{i-1}}{U_{(i)} - U_{(i-1)}} | U_{(i-1)}, U_{(i)} \right] = r^{(1)}(U_{(i)}) + o_p(1)$$

and

$$\mathbf{Var}[\hat{q}_i^{(1)} | U_{(i-1)}, U_{(i)}] = \mathbf{Var}\left[ \frac{Y_i - Y_{i-1}}{U_{(i)} - U_{(i-1)}} | U_{(i-1)}, U_{(i)} \right] = O_p(n^2).$$

It is immediately clear that the first order difference quotient proposed by [16] is an asymptotic unbiased estimator of $r(U_{(i)})$. The variance of this estimator can be arbitrary large, severely complicating derivative estimation and the main goal to be addressed in this paper.

## 2.2 Asymptotic properties of the first order derivative estimator

The following theorems establish the asymptotic bias and variance of our proposed estimator (5) for interior points i.e., $k + 1 \leq i \leq n - k$. In what follows we denote $\mathbb{U} = (U_{(i-j)}, \ldots, U_{(i+j)})$ for $i > j$ and $j = 1, \ldots, k$.

**Theorem 1** *Under model* (4) *and assume* $r(\cdot)$ *is twice continuously differentiable on* $[0, 1]$ *and* $k \to \infty$ *as* $n \to \infty$. *Then, for uniform random design on* $[0, 1]$ *and for the weights in Proposition 1, the conditional (absolute) bias and conditional variance of* (5) *are given by*

$$\left| \text{bias}\big[\hat{Y}_i^{(1)} | \mathbb{U}\big] \right| \leq \sup_{u \in [0,1]} |r^{(2)}(u)| \frac{3k(k+1)}{4(n+1)(2k+1)} + o_p(n^{-1}k)$$

*and*

$$\mathbf{Var}\big[\hat{Y}_i^{(1)} | \mathbb{U}\big] = \frac{3\sigma_e^2(n+1)^2}{k(k+1)(2k+1)} + o_p(n^2 k^{-3})$$

*uniformly for* $k + 1 \leq i \leq n - k$.

From Theorem 1, the pointwise consistency immediately follows.

**Corollary 1** *Under the assumptions of Theorem 1,* $k \to \infty$ *as* $n \to \infty$ *such that* $n^{-1}k \to 0$ *and* $n^2 k^{-3} \to 0$. *Then, for* $\sigma_e^2 < \infty$ *and the weights given in Proposition 1, we have for any* $\varepsilon > 0$

$$\mathbf{P}(|\hat{Y}_i^{(1)} - r^{(1)}(U_{(i)})| \geq \varepsilon) \to 0$$

*for* $k + 1 \leq i \leq n - k$.

According to Theorem 1 and Corollary 1, the conditional bias and variance of (5) goes to zero as $n \to \infty$ and $k \to \infty$ faster than $O(n^{2/3})$, but slower than $O(n)$. In the next section, we propose a selection method for $k$ in practice such that $k = O(n^{4/5})$. The fastest possible rate at which $\mathbf{E}[(\hat{Y}_i^{(1)} - r^{(1)}(U_{(i)}))^2 | \mathbb{U}] \to 0$ ($L_2$ rate of convergence) is $O_p(n^{-2/5})$. Using Jensen's inequality, similar results can be shown for the $L_1$ rate of convergence i.e.,

$$\mathbf{E}\big[|\hat{Y}_i^{(1)} - r^{(1)}(U_{(i)})| \mid \mathbb{U}\big] \leq \left| \text{bias}\big[\hat{Y}_i^{(1)} | \mathbb{U}\big] \right| + \sqrt{\mathbf{Var}\big[\hat{Y}_i^{(1)} | \mathbb{U}\big]} = O_p(n^{-1/5}).$$

## 2.3 Selection method for $k$

Crucial to the estimator (5) is the parameter $k$ which controls the bias-variance trade-off. Based on Theorem 1, we choose the $k$ that minimizes the asymptotic upper bound of the mean integrated squared error (MISE). The result is given in Corollary 2. However, a closed form for $k_{\text{opt}}$ cannot be obtained.

**Corollary 2** *Under the assumptions of Theorem 1 and denote* $\mathcal{B} = \sup_{u \in [0,1]} |r^{(2)}(u)|$, *then the* $k$ *that minimizes asymptotic upper bound of MISE is*

$$k_{\text{opt}} = \arg\min_{k \in \mathbb{N}^+ \setminus \{0\}} \left\{ \mathcal{B}^2 \frac{9k^2(k+1)^2}{16(n+1)^2(2k+1)^2} + \frac{3\sigma_e^2(n+1)^2}{k(k+1)(2k+1)} \right\} = O(n^{4/5}).$$

The only unknown two quantities here are $\sigma_e^2$ and $\mathcal{B}$. The error variance can be estimated by Hall's $\sqrt{n}$-consistent estimator [26]

$$\hat{\sigma}_e^2 = \frac{1}{n-2} \sum_{i=1}^{n-2} (0.809Y_i - 0.5Y_{i+1} - 0.309Y_{i+2})^2.$$

The second unknown quantity $\mathcal{B}$ can be (roughly) estimated with a local polynomial regression estimator of order $p = 3$. The performance of our proposed model is not so sensitive to the accuracy of $\mathcal{B}$, thus a rough estimate of the second order derivative is sufficient. By plugging in these two estimators for $\sigma_e^2$ and $\mathcal{B}$ in Corollary 2, the optimal value $k_{\text{opt}}$ can be obtained using a grid search (or any other optimization method) over the integer set $[1, \lceil \frac{n-1}{2} \rceil]$.

## 2.4 Asymptotic order of the bias and continuous differentiability of $r$

In Theorem 1, we bounded the conditional bias above. From a theoretical point of view, it is helpful to derive an exact expression for the conditional bias. Assume that the first $q + 1$ derivatives of $r(\cdot)$ exist on $[0, 1]$. A Taylor series of $r(U_{(i\pm j)})$ in a neighborhood of $U_{(i)}$ yields

$$r(U_{(i\pm j)}) \quad = \quad r(U_{(i)}) + \sum_{l=1}^{q} \frac{1}{l!}(U_{(i\pm j)} - U_{(i)})^l r^{(l)}(U_{(i)}) + O_p\{(j/n)^{q+1}\}.$$

Using Lemma 1, assume $k \to \infty$ as $n \to \infty$, and for the weights in Proposition 1 we obtain the asymptotic order of the exact conditional bias for different values of $q$

$$\text{bias}\big[\hat{Y}_i^{(1)}|\mathbb{U}\big] = \begin{cases} O_p\left(\frac{k}{n}\right), & q = 1; \\ O_p\{\max\left(\frac{k^{\frac{1}{2}}}{n}, \frac{k^2}{n^2}\right)\}, & q \geq 2. \end{cases}$$

For $q = 1$ (i.e., $r(\cdot)$ is twice continuously differentiable), the leading order of exact conditional bias is the same as that of the bias upperbound given in Theorem 1. For $q = 2$, $r(\cdot)$ is three times continuously differentiable on $[0, 1]$ and the exact bias achieves smaller order than $O_p(k/n)$. Adding additional assumptions on the differentiability of $r(\cdot)$, i.e. $q > 2$, will no longer improve the asymptotic rate of the bias. This can be seen as follows: for $q \geq 2$, the bias is

$$\text{bias}\big[\hat{Y}_i^{(1)}|\mathbb{U}\big] \quad = \quad \frac{\sum_{j=1}^{k}(U_{(i+j)} - U_{(i-j)})\big[\sum_{l=2}^{q} \frac{r^{(l)}(U_{(i)})\{(U_{(i+j)}-U_{(i)})^l - (U_{(i-j)}-U_{(i)})^l\}}{l!} + O_p\{(j/n)^{q+1}\}\big]}{\sum_{p=1}^{k}(U_{(i+p)} - U_{(i-p)})^2}.$$

This can be split into two terms: odd and even $l \geq 2$

$$\text{bias}_{\text{odd}}[\hat{Y}_i^{(1)} \mid \mathbb{U}] = O_p\left(\frac{k^2}{n^2}\right) \qquad \text{and} \qquad \text{bias}_{\text{even}}[\hat{Y}_i^{(1)} \mid \mathbb{U}] = O_p\left(\frac{k^{\frac{1}{2}}}{n}\right)$$

resulting in

$$\begin{aligned} \text{bias}\big[\hat{Y}_{[i]}^{(1)} \mid \mathbb{U}\big] \quad &= \quad \text{bias}_{\text{odd}}[\hat{Y}_i^{(1)} \mid \mathbb{U}] + \text{bias}_{\text{even}}[\hat{Y}_i^{(1)} \mid \mathbb{U}] \\ &= \quad O_p\left\{\max\left(\frac{k^2}{n^2}, \frac{k^{\frac{1}{2}}}{n}\right)\right\}. \end{aligned}$$

In fixed design, $\text{bias}_{\text{even}} = 0$ due to symmetry: $u_{(i+j)} - u_{(i)} = u_{(i)} - u_{(i-j)}$. Unfortunately, in the random design, we cannot remove $\text{bias}_{\text{even}}$. It is this fact that will lead to the inconsistency of third and higher order derivatives if these estimators are defined in a fully recursive way as in [21]. Due to page limitations we will not elaborate further on higher order derivative estimation, but more information and theoretical results can be obtained by contacting the first author.

## 2.5 Boundary Correction

We already discussed the proposed estimator at the interior points and in this section we provide a simple but effective correction for the boundary region. Points with index $i < k + 1$ and $i > n - k$ are points located at the left and right boundary respectively. Since there are not enough $k$ pairs of neighbors at the boundary, we use a weighted linear combination of $k(i)$ pairs at points $U_i$ instead, where $k(i) = i - 1$ for the left boundary and $k(i) = n - i$ for the right boundary. The first order derivative estimator at the boundary is obtained by replacing $k$ with $k(i)$ in the estimator (5) and weights in Proposition 1. From Section 2.4, assume $r(\cdot)$ is three times continuously differentiable on $[0, 1]$, at the boundary, the asymptotic order of the bias is $O_p\{\max\left(\frac{k(i)^2}{n^2}, \frac{k(i)^{1/2}}{n}\right)\}$, which is smaller than the interior points. However, the asymptotic order of the variance is $O_p\{(3\sigma_e^2(n + 1)^2)/(k(i)(k(i) + 1)(2k(i) + 1))\}$ and attains $O_p(n^2)$, as $i$ is close to either 2 or $n - 1$.

In order to reduce the variance at the boundary we propose the following modification to (5). For points at the left boundary, $i < k + 1$, the estimator becomes

$$\hat{Y}_i^{(1)} \quad = \quad \sum_{j=1}^{k(i)} w_{i,j} \cdot \left(\frac{Y_{i+j} - Y_{i-j}}{U_{(i+j)} - U_{(i-j)}}\right) + \sum_{j=k(i)+1}^{k} w_{i,j} \cdot \left(\frac{Y_{i+j} - Y_i}{U_{(i+j)} - U_{(i)}}\right) \qquad (8)$$

with

$$w_{i,j} = \begin{cases} \dfrac{(U_{(i+j)} - U_{(i-j)})^2}{\sum_{l=1}^{k(i)}(U_{(i+l)} - U_{(i-l)})^2 + \sum_{l=k(i)+1}^{k}(U_{(i+l)} - U_{(i)})^2}, & 1 \leq j \leq k(i); \\[2em] \dfrac{(U_{(i+j)} - U_{(i)})^2}{\sum_{l=1}^{k(i)}(U_{(i+l)} - U_{(i-l)})^2 + \sum_{l=k(i)+1}^{k}(U_{(i+l)} - U_{(i)})^2}, & k(i) < j \leq k. \end{cases}$$

This modification leads to

$$\text{bias}[\hat{Y}_i^{(1)}|\mathbb{U}] = O_p\left\{\max\left(\frac{k(i)^{7/2}}{k^3 n}, \frac{k(i)^5}{k^3 n^2}, \frac{k - k(i)}{n}\right)\right\}$$

and

$$\mathbf{Var}[\hat{Y}_i^{(1)}|\mathbb{U}] = O_p\left\{\max\left(\frac{n^2}{k^3}, \frac{n^2(k - k(i))^2}{k^4}\right)\right\}.$$

The $\text{bias}[\hat{Y}_i^{(1)}|\mathbb{U}] \to 0$ when $n \to \infty$ indicating that (8) is still asymptotically unbiased at the boundary. Worst case scenario, the variance is of the order $O_p(n^2/k^2)$ which is smaller than $O_p(n^2)$. A similar result can be obtained for the right boundary.

## 2.6 Smoothed first order derivative estimation

The estimators (5) and (8) generate first order derivatives which still contain the noise coming from the unknown errors $e_i, i = 1, \ldots, n$ in model (4) and can only be evaluated at the design points $U_{(i)}, i = 1, \ldots, n$. In order to evaluate the derivative in an arbitrary point we propose smoothing the newly generated data set. However, from (5) it is clear that for the generated derivatives $\hat{Y}_i^{(1)}, i = 1, \ldots, n$ the i.i.d. assumption is no longer valid since it is a weighted sum of differences of the original data set. Hence, bandwidth selection for any nonparametric smoothing method becomes increasingly difficult. We rewrite estimator (5) in the form of the smoothed first order derivative

$$\hat{Y}_i^{(1)} = r_2^{(1)}(U_{(i)}) + \varepsilon_i, \qquad i = 1, \ldots, n$$

where $\hat{Y}_i^{(1)}$ is the first order derivative, given by (5), $r_2^{(1)}(U_{(i)}) = r^{(1)}(U_{(i)}) + \text{bias}[\hat{Y}_i^{(1)}|\mathbb{U}]$ where $r^{(1)}(\cdot)$ is the smoothed (and our final) first order derivative estimate and $\varepsilon_i = \sum_{j=1}^{k} w_{i,j} \cdot \left(\frac{e_{i+j} - e_{i-j}}{U_{(i+j)} - U_{(i-j)}}\right)$. Based on model (4), $\mathbf{E}[\varepsilon|U] = 0$ and $\mathbf{Cov}(\varepsilon_i, \varepsilon_j|U_{(i)}, U_{(j)}) = \sigma_e^2 \rho_n(U_{(i)} - U_{(j)})$ for $i \neq j$ with $\sigma_e^2 < \infty$ and $\rho_n$ is a stationary correlation function satisfying $\rho_n(0) = 1, \rho_n(u) = \rho_n(-u)$ and $|\rho_n(u)| \leq 1$ for all $u$. The subscript $n$ allows the correlation function $\rho_n$ to shrink as $n \to \infty$ [27]. Under mild assumptions on the correlation function, which is unknown, [27] showed that, by using a bimodal type of kernel $K$ such that $K(0) = 0$, the residual sum of squares (RSS) approximates the asymptotic squared error uniformly over a set of bandwidths. Consequently, choosing the bandwidth $\hat{h}_b$ (of the bimodal kernel) minimizing the RSS results in an optimal bandwidth that minimizes the asymptotic squared error asymptotically. As bimodal kernels introduce extra error in the estimation due to their non-optimality we overcome this issue by using $\hat{h}_b$ as a pilot bandwidth and relate it to a bandwidth $\hat{h}$ of a more optimal (unimodal) kernel, say the Gaussian kernel. As shown in [27], this can be achieved without any extra smoothing step. For local cubic regression, the relation between the bimodal and unimodal bandwidth is

$$\hat{h} = 1.01431\hat{h}_b$$

when using $\overline{K}(u) = (2/\sqrt{\pi})u^2 \exp(-u^2)$ and $K(u) = (1/\sqrt{2\pi})\exp(-u^2/2)$ as bimodal and unimodal kernel respectively. Following the proof of [1, p. 101-103], it can be shown that $\hat{r}_2^{(1)}(\cdot)$ is a consistent estimator of $r^{(1)}(\cdot)$. In what follows, denote $\hat{r}_2^{(1)}(\cdot)$ by $\hat{r}^{(1)}(\cdot)$.

## 2.7 Generalizing first order derivatives to any continuous distribution

It is possible to find a closed form expression for the distribution of the differences $X_{(i+j)} - X_{(i-j)}$ with $X \overset{i.i.d}{\sim} F$ where $F$ is unknown and continuous [25] such that the density function $f(x) = F'(x)$.

Since this result is quite unattractive from a theoretical point of view, we advocate the use of the probability integral transform stating that

$$F(X) \sim U(0,1). \tag{9}$$

By using the probability integral transform we know that the new data set $(F(X_{(1)}), Y_1), \ldots, (F(X_{(n)}), Y_n)$ is the same as $(U_{(1)}, Y_1), \ldots, (U_{(n)}, Y_n)$. This leads back to the original setting of uniform order statistics discussed earlier. The final step is to transform back to the original space. In order for this step to work we need the existence of a density $f$. Since $m(X) = r(F(X))$ and by the chain rule

$$\frac{dm(X)}{dX} = \frac{dr(U)}{dU}\frac{dU}{dX} = f(X)\frac{dr(U)}{dU} \tag{10}$$

yielding $m^{(1)}(X) = f(X)r^{(1)}(U)$ which is the smoothed version of the first order derivative in the original space. In practice the distribution $F$ and density $f$ need to be estimated giving $\widehat{m}^{(1)}(X) = \widehat{f}(X)\widehat{r}^{(1)}(U)$. In this paper we use the kernel density estimator [28, 29] to estimate the density $f$ and distribution $F$.

## 3  Simulations

Consider the following function

$$m(X) = \cos^2(2\pi X) \quad \text{for} \quad X \sim \text{beta}(2,2), \tag{11}$$

with sample size $n = 700$ and $e \sim N(0, 0.2^2)$. We pretend we do not know the underlying distribution of $X$ in model 11, since this is what occurs in applications. We use the kernel density estimator [30] to estimate the density $f$ and cumulative distribution $F$. The tuning parameter $k$ is selected over the integer set $[1, \lceil (n-1)/2 \rceil]$ and according to Corollary 2. We use local cubic regression ($p = 3$) with bimodal kernel to initially smooth the data. Bandwidths for the bimodal kernel $\hat{h}_b$ are selected from the set $\{0.1, 0.105, 0.11, \ldots, 0.2\}$ and corrected for a unimodal Gaussian kernel. Figure 1a shows the first order noisy derivative (blue dots) and smoothed first order derivatives (dashed line) of $r(\cdot)$ after using the probability integral transform. Using (10), the smoothed first order derivative $\hat{m}^{(1)}(\cdot)$ (dashed line) in the original space is shown in Figure 1b. Figure 1b also shows the true first order derivative (full line) and the derivatives estimated by local quadratic regression [31] (dash-dotted line) for comparison purposes. Compared to the local polynomial derivative estimator in Figure 1b, the proposed estimator is slightly better in the interior for model (11). However, both methods suffer from boundary effects. Next, we compare the proposed methodology to several popular methods for nonparametric derivative estimation, i.e. the local slope of the local polynomial regression estimator with $p = 2$ and penalized cubic smoothing splines [32]. The order of the local polynomial is set to $p = 2$, since $p$ minus the order of the derivative is odd [1], and penalized smoothing cubic splines are used for the spline derivative estimator. For the Monte Carlo study, we constructed data sets of size $n = 700$ and generated the functions

$$m(X) = \sqrt{X(1-X)}\sin\{(2.1\pi)/(X + 0.05)\} \quad \text{for} \quad X \sim \mathcal{U}(0.25, 1) \tag{12}$$

and

$$m(X) = \sin(2X) + 2\exp(-16X^2) \quad \text{for} \quad X \sim \mathcal{U}(-1, 1) \tag{13}$$

100 times according to model (2) with $e \sim N(0, 0.2^2)$ and $e \sim N(0, 0.3^2)$ for model (12) and model (13) respectively. Bandwidths are selected from the set $\{0.04, 0.045, \ldots, 0.08\}$ and corrected for a unimodal Gaussian kernel. In order to remove boundary effects for all three methods, we use the adjusted mean absolute error as a performance measure which we define as

$$\text{MAEadjusted} = \frac{1}{670}\sum_{i=16}^{685}|\widehat{m}'(X_i) - m'(X_i)|.$$

The result is shown in Figure 2. The proposed method loses some performance due to estimating $f$ and $F$. If $F(X)$ and $f(X)$ are known, the proposed method will have a better performance. In general, the proposed method has equal performance to local quadratic regression and cubic penalized smoothing splines.

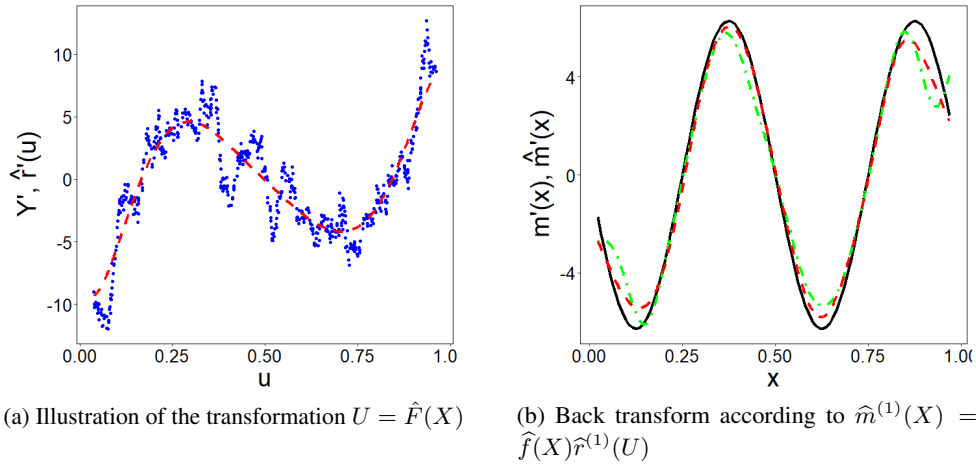

(a) Illustration of the transformation $U = \hat{F}(X)$

(b) Back transform according to $\widehat{m}^{(1)}(X) = \widehat{f}(X)\widehat{r}^{(1)}(U)$

Figure 1: Illustration of the proposed methodology: (a) First order noisy derivative (dots), after probability integral transform of original data, of model (11) based on $k = 22$, smoothed derivative based on local cubic regression (dashed line); (b) true derivative (full line), smoothed derivative based on local cubic regression (dashed line) and local polynomial derivative with $p = 2$ (dash-dotted line) in the original space. Boundary points are not shown for visual purposes.

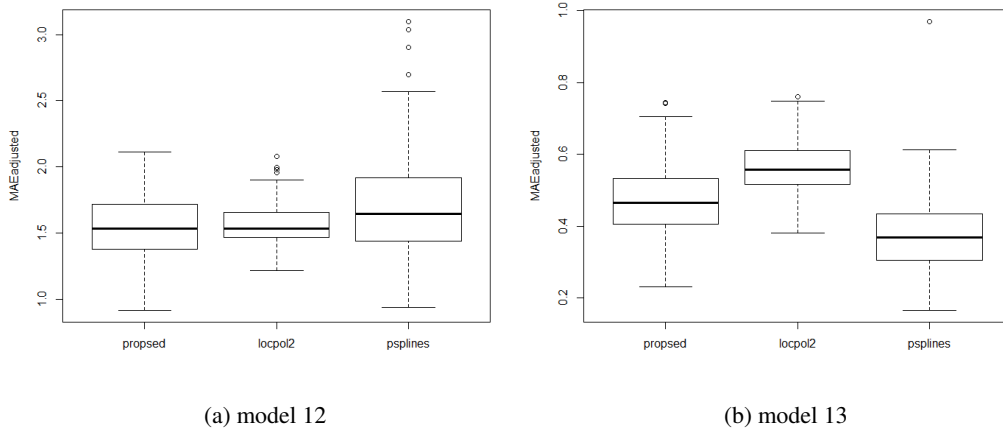

(a) model 12

(b) model 13

Figure 2: Results of the Monte Carlo study for the proposed methodology, local quadratic regression and penalized smoothing splines for first order derivative estimation.

## 4    Conclusions

In this paper we proposed a theoretical framework for first order derivative estimation based on a variance-reducing linear combination of symmetric quotients for random design. Although this is a popular estimator for the equispaced design case, we showed that for the random design some difficulties occur and extra estimation of unknown quantities are needed. It is also possible to extend these type of estimators to higher order derivatives and similar theoretical results can be established.

## Footnotes

*Liu and De Brabanter are with Iowa State University, Ames, IA 50011, USA. Corresponding authors:

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
