[Supplementary Material · supplementary material.pdf]

# Supplementary Material for Derivative Estimation in Random Design

**Yu Liu[1], Kris De Brabanter[1,2*]**

[1]Department of Computer Science, [2]Department of Statistics

## 1  Proof of Proposition 1

$$
\begin{aligned}
\mathbf{Var}\big[\hat{Y}_i^{(1)}|\mathbb{U}\big] &= \mathbf{Var}\left[\sum_{j=1}^{k} w_{i,j} \cdot \left(\frac{Y_{i+j}-Y_{i-j}}{U_{(i+j)}-U_{(i-j)}}\right) \mid \mathbb{U}\right] \\
&= \left(1-\sum_{j=2}^{k} w_{i,j}\right)^2 \mathbf{Var}\left[\frac{Y_{i+1}-Y_{i-1}}{U_{(i+1)}-U_{(i-1)}} \mid \mathbb{U}\right] \\
&\quad + \sum_{j=2}^{k} w_{i,j}^2 \, \mathbf{Var}\left[\frac{Y_{i+j}-Y_{i-j}}{U_{(i+j)}-U_{(i-j)}} \mid \mathbb{U}\right] \\
&= \left(1-\sum_{j=2}^{k} w_{i,j}\right)^2 \frac{2\sigma_e^2}{(U_{(i+1)}-U_{(i-1)})^2} + \frac{2\sigma_e^2}{(U_{(i+j)}-U_{(i-j)})^2}\sum_{j=2}^{k} w_{i,j}^2 .
\end{aligned}
$$

Setting the partial derivatives to zero yields

$$
w_{i,j} = w_{i,1} \frac{(U_{(i+j)}-U_{(i-j)})^2}{(U_{(i+1)}-U_{(i-1)})^2}.
$$

Using the fact that $\sum_{j=1}^{k} w_{i,j} = 1$ results in

$$
\sum_{j=1}^{k} w_{i,j} = w_{i,1}\sum_{j=1}^{k}\frac{(U_{(i+j)}-U_{(i-j)})^2}{(U_{(i+1)}-U_{(i-1)})^2} = 1.
$$

Consequently, this gives

$$
w_{i,j}\frac{(U_{(i+1)}-U_{(i-1)})^2}{(U_{(i+j)}-U_{(i-j)})^2}\sum_{j=1}^{k}\frac{(U_{(i+j)}-U_{(i-j)})^2}{(U_{(i+1)}-U_{(i-1)})^2} = 1
$$

proving the proposition.

## 2  Proof of Lemma 1

Following [1, p. 14] we have

$$
U_{(i+j)} - U_{(i-j)} \sim \text{Beta}(2j, n+1-2j).
$$

It immediately follows that

$$
\begin{aligned}
U_{(i+j)} - U_{(i-j)} &= \mathbf{E}\{U_{(i+j)} - U_{(i-j)}\} + O_p\left[\sqrt{\mathbf{Var}\{U_{(i+j)} - U_{(i-j)}\}}\right] \\
&= \frac{2j}{n+1} + O_p\left(\sqrt{\frac{j}{n^2}}\right)
\end{aligned}
$$

Similarly, according to the property of uniform order statistics we have

$$
U_{(i+j)} - U_{(i)} \sim \text{Beta}(j, n+1-j)
$$

and

$$
\begin{aligned}
U_{(i+j)} - U_{(i)} &= \mathbf{E}\{U_{(i+j)} - U_{(i)}\} + O_p\left[\sqrt{\mathbf{Var}\{U_{(i+j)} - U_{(i)}\}}\right] \\
&= \frac{j}{n+1} + O_p\left(\sqrt{\frac{j}{n^2}}\right).
\end{aligned}
$$

The proof of the third part of the lemma is analogous to the proof above and is therefore omitted.

## 3   Proof of Theorem 1

Since $r(\cdot)$ is twice continuously differentiable on $[0,1]$, the following Taylor expansions are valid for $r(U_{(i+j)})$ and $r(U_{(i-j)})$ in a neighborhood of $U_{(i)}$:

$$
r(U_{(i+j)}) = r(U_{(i)}) + (U_{(i+j)} - U_{(i)})r'(U_{(i)}) + \frac{(U_{(i+j)} - U_{(i)})^2}{2}r^{(2)}(\zeta_{i,i+j})
$$

and

$$
r(U_{(i-j)}) = r(U_{(i)}) + (U_{(i-j)} - U_{(i)})r'(U_{(i)}) + \frac{(U_{(i-j)} - U_{(i)})^2}{2}r^{(2)}(\zeta_{i-j,i}),
$$

where $\zeta_{i,i+j} \in ]U_{(i)}, U_{(i+j)}[$ and $\zeta_{i-j,i} \in ]U_{(i-j)}, U_{(i)}[$. Using Lemma 1 and Proposition 1, the absolute conditional bias is bounded above by

$$
\begin{aligned}
\left|\text{bias}\big[\hat{Y}_i^{(1)} \mid \mathbb{U}\big]\right| &= \left|\mathbf{E}\left[\sum_{j=1}^k w_{i,j} \cdot \left(\frac{Y_{i+j} - Y_{i-j}}{U_{(i+j)} - U_{(i-j)}}\right) \mid \mathbb{U}\right] - r'(U_i)\right| \\
&= \frac{1}{2}\left|\sum_{j=1}^k w_{i,j} \frac{(U_{(i+j)} - U_{(i)})^2 r^{(2)}(\zeta_{i,i+j}) - (U_{(i-j)} - U_{(i)})^2 r^{(2)}(\zeta_{i-j,i})}{U_{(i+j)} - U_{(i-j)}}\right| \\
&= \frac{1}{2}\left|\frac{1}{\sum_{l=1}^k (U_{(i+l)} - U_{(i-l)})^2}\left(\sum_{j=1}^k (U_{(i+j)} - U_{(i-j)})\{(U_{(i+j)} - U_{(i)})^2 r^{(2)}(\zeta_{i,i+j})\right.\right. \\
&\qquad\left.\left. - (U_{(i-j)} - U_{(i)})^2 r^{(2)}(\zeta_{i-j,i})\}\right)\right| \\
&\le \frac{1}{2}\sup_{u\in[0,1]}|r^{(2)}(u)|\frac{\sum_{j=1}^k (U_{(i+j)} - U_{(i-j)})\{(U_{(i+j)} - U_{(i)})^2 + (U_{(i-j)} - U_{(i)})^2\}}{\sum_{l=1}^k (U_{(i+l)} - U_{(i-l)})^2} \\
&= \frac{1}{2}\sup_{u\in[0,1]}|r^{(2)}(u)|\frac{\frac{k^2(k+1)^2}{(n+1)^3}\{1 + O_p(\frac{1}{\sqrt{k}})\}}{\frac{2k(k+1)(2k+1)}{3(n+1)^2}\{1 + O_p(\frac{1}{\sqrt{k}})\}} \\
&= \sup_{u\in[0,1]}|r^{(2)}(u)|\frac{3k(k+1)}{4(n+1)(2k+1)}\left\{1 + O_p\left(\frac{1}{\sqrt{k}}\right)\right\}.
\end{aligned}
$$

Then for $k \to \infty$ as $n \to \infty$

$$
\left|\text{bias}\big[\hat{Y}_i^{(1)} \mid \mathbb{U}\big]\right| \le \sup_{u\in[0,1]}|r^{(2)}(u)|\frac{3k(k+1)}{4(n+1)(2k+1)}\{1 + o_p(1)\}.
$$

Using Proposition 1, the conditional variance yields

$$
\begin{aligned}
\mathbf{Var}\big[\hat{Y}_i^{(1)}|\mathbb{U}\big] &= \mathbf{Var}\left[\sum_{j=1}^{k} w_{i,j}\cdot\left(\frac{Y_{i+j}-Y_{i-j}}{U_{(i+j)}-U_{(i-j)}}\right)\,\Big|\,\mathbb{U}\right] \\
&= 2\sigma_e^2\,\frac{\sum_{j=1}^{k}(U_{(i+j)}-U_{(i-j)})^2}{\left(\sum_{l=1}^{k}(U_{(i+l)}-U_{(i-l)})^2\right)^2} \\
&= 2\sigma_e^2\,\frac{1}{\sum_{l=1}^{k}\left(U_{(i+l)}-U_{(i-l)}\right)^2} \\
&= 2\sigma_e^2\,\frac{1}{\frac{2k(k+1)(2k+1)}{3(n+1)^2}\,\{1+o_p(1)\}} \\
&= \frac{3\sigma_e^2(n+1)^2}{k(k+1)(2k+1)}\,\{1+o_p(1)\},
\end{aligned}
$$

provided that $k \to \infty$ as $n \to \infty$. Both results hold uniformly for $k+1 \le i \le n-k$.

## 4  Proof of Corollary 1

Under the conditions $k \to \infty$ as $n \to \infty$ such that $n^{-1}k \to 0$ and $n^2 k^{-3} \to 0$, Theorem 1 states that the upperbound of conditional bias and conditional variance go to zero. Consequently, we have that

$$
\lim_{n\to\infty}\mathrm{MSE}\big[\hat{Y}_i^{(1)}|\mathbb{U}\big] = 0.
$$

According to Chebyshev's inequality the proof is complete.

## 5  Proof of Corollary 2

From the bias-variance decomposition of the mean squared error (MSE), it follows that

$$
\mathrm{MSE}\big[\hat{Y}_i^{(1)}|\mathbb{U}\big] \le \mathcal{B}^2\frac{9k^2(k+1)^2}{16(n+1)^2(2k+1)^2} + \frac{3\sigma_e^2(n+1)^2}{k(k+1)(2k+1)} + o_p(n^{-2}k^2+n^2k^{-3}),
$$

with $\mathcal{B}=\sup_{u\in[0,1]}|r^{(2)}(u)|$. Since $U \sim \mathcal{U}(0,1)$, the mean integrated squared error (MISE) which measures the average global error is

$$
\begin{aligned}
\mathrm{MISE}\big[\hat{Y}^{(1)}|\mathbb{U}\big] &= \int_0^1 \mathrm{MSE}\big[\hat{Y}_i^{(1)}|\mathbb{U}\big]\,du \\
&\le B^2\frac{9k^2(k+1)^2}{16(n+1)^2(2k+1)^2} + \frac{3\sigma_e^2(n+1)^2}{k(k+1)(2k+1)} + o_p(n^{-2}k^2+n^2k^{-3}).
\end{aligned}
$$

Denote the asymptotic MISE (AMISE) by

$$
\mathrm{AMISE}\big[\hat{Y}^{(1)}|\mathbb{U}\big] \le \mathcal{B}^2\frac{9k^2(k+1)^2}{16(n+1)^2(2k+1)^2} + \frac{3\sigma_e^2(n+1)^2}{k(k+1)(2k+1)}.
$$

# 6 Asymptotic order of the bias and continuous differentiability of $r$

Assume the $q + 1$ derivatives of $r(\cdot)$ exist on $[0, 1]$, the following Taylor expansions are valid for $r(U_{(i+j)})$ and $r(U_{(i-j)})$ in a neighborhood of $U_{(i)}$

$$
\begin{aligned}
r(U_{(i+j)}) &= r(U_{(i)}) + \sum_{l=1}^{q} \frac{1}{l!}(U_{(i+j)} - U_{(i)})^l r^{(l)}(U_{(i)}) + O_p(U_{(i+j)} - U_{(i)})^{q+1} \\
&= r(U_{(i)}) + \sum_{l=1}^{q} \frac{1}{l!}(U_{(i+j)} - U_{(i)})^l r^{(l)}(U_{(i)}) + O_p\{(j/n)^{q+1}\} \\
r(U_{(i-j)}) &= r(U_{(i)}) + \sum_{l=1}^{q} \frac{1}{l!}(U_{(i-j)} - U_{(i)})^l r^{(l)}(U_{(i)}) + O_p(U_{(i-j)} - U_{(i)})^{q+1} \\
&= r(U_{(i)}) + \sum_{l=1}^{q} \frac{1}{l!}(U_{(i-j)} - U_{(i)})^l r^{(l)}(U_{(i)}) + O_p\{(j/n)^{q+1}\}.
\end{aligned}
$$

Taking expectations, using Lemma 1 and for $\sum_{j=1}^{k} w_{i,j} = 1$

$$
\begin{aligned}
\mathbf{E}\big[\hat{Y}_i^{(1)}|\mathbb{U}\big] &= \sum_{j=1}^{k} w_{i,j} \frac{r(U_{(i+j)}) - r(U_{(i-j)})}{U_{(i+j)} - U_{(i-j)}} \\
&= \frac{1}{\sum_{p=1}^{k}(U_{(i+p)} - U_{(i-p)})^2} \Bigg( \sum_{j=1}^{k}(U_{(i+j)} - U_{(i-j)})\big[\sum_{l=1}^{q} \frac{r^{(l)}(U_{(i)})}{l!}\big\{(U_{(i+j)} - U_{(i)})^l \\
&\quad - (U_{(i-j)} - U_{(i)})^l\big\} + O_p\{(j/n)^{q+1}\}\big]\Bigg)
\end{aligned}
$$

For $q = 1$

$$
\begin{aligned}
\text{bias}\big[\hat{Y}_i^{(1)}|\mathbb{U}\big] &= \frac{r^{(1)}(U_{(i)})\sum_{j=1}^{k}(U_{(i+j)} - U_{(i-j)})^2 + O_p\left(k^4/n^3\right)}{\sum_{p=1}^{k}(U_{(i+p)} - U_{(i-p)})^2} - r^{(1)}(U_{(i)}) \\
&= O_p\left(\frac{k}{n}\right)
\end{aligned}
$$

and for $q = 2$

$$
\begin{aligned}
\text{bias}\big[\hat{Y}_i^{(1)}|\mathbb{U}\big] &= \frac{r^{(2)}(U_{(i)})\sum_{j=1}^{k}(U_{(i+j)} - U_{(i-j)})\big\{(U_{(i+j)} - U_{(i)})^2 - (U_{(i-j)} - U_{(i)})^2\big\}}{2\sum_{p=1}^{k}(U_{(i+p)} - U_{(i-p)})^2} \\
&\quad + \frac{O_p\left(k^5/n^4\right)}{2\sum_{p=1}^{k}(U_{(i+p)} - U_{(i-p)})^2} \\
&= \frac{O_p(k^{\frac{7}{2}}/n^3) + O_p\left(k^5/n^4\right)}{O_p(k^3/n^2)} \\
&= O_p\left\{\max\left(\frac{k^{\frac{1}{2}}}{n}, \frac{k^2}{n^2}\right)\right\}.
\end{aligned}
$$

The bias can be split into two terms, $\text{bias}_{\text{even}} = O_p(k^{1/2}/n)$ and $\text{bias}_{\text{odd}} = O_p(k^2/n^2)$. $\text{bias}_{\text{even}}$ indicates the bias contribution from the even order term in the Taylor expansion of $r(U_{(i\pm j)})$ and $\text{bias}_{\text{odd}}$ for the odd order term. An analogous result can be obtained for $q > 2$.

## 7   Bias and Variance at the Left Boundary

Assume that $r(\cdot)$ is three times continuously differentiable on $[0, 1]$. At the left boundary $i < k + 1$, we have

$$
\begin{aligned}
\text{bias}[\hat{Y}_i^{(1)}|\mathbb{U}] &= \sum_{j=1}^{k(i)} w_{i,j} \cdot \left( \frac{\frac{1}{2}\left[ (U_{(i+j)} - U_{(i)})^2 - \frac{1}{2}(U_{(i-j)} - U_{(i)})^2 \right] r^{(2)}(U_{(i)})}{U_{(i+j)} - U_{(i-j)}} \right) \\
&+ \sum_{j=1}^{k(i)} w_{i,j} \cdot \left( \frac{O_p(j^3/n^3)}{U_{(i+j)} - U_{(i-j)}} \right) \\
&+ \sum_{j=k(i)+1}^{k} w_{i,j} \cdot \left( \frac{1}{2}(U_{(i+j)} - U_{(i)})r^{(2)}(U_{(i)}) \right) \left\{ 1 + o_p(1) \right\} \\
&= O_p\left\{ \max\left( \frac{k(i)^{7/2}}{k^3 n}, \frac{k(i)^5}{k^3 n^2}, \frac{k - k(i)}{n} \right) \right\}
\end{aligned}
$$

$$
\begin{aligned}
\textbf{Var}[\hat{Y}_i^{(1)}|\mathbb{U}] &= \textbf{Var}\left[ \sum_{j=1}^{k(i)} w_{i,j} \left( \frac{Y_{i+j} - Y_{i-j}}{U_{(i+j)} - U_{(i-j)}} \right) |\mathbb{U} \right] + \textbf{Var}\left[ \sum_{j=k(i)+1}^{k} w_{i,j} \left( \frac{Y_{i+j} - Y_i}{U_{(i+j)} - U_{(i)}} \right) |\mathbb{U} \right] \\
&= 2\sigma_e^2 \sum_{j=1}^{k(i)} \left( \frac{w_{i,j}}{U_{(i+j)} - U_{(i-j)}} \right)^2 + \sigma_e^2 \sum_{j=k(i)+1}^{k} \left( \frac{w_{i,j}}{U_{(i+j)} - U_{(i)}} \right)^2 \\
&+ \sigma_e^2 \sum_{j=k(i)+1}^{k} \sum_{l=k(i)+1}^{k} \left( \frac{w_{i,j}}{U_{(i+j)} - U_{(i)}} \right) \left( \frac{w_{i,l}}{U_{(i+l)} - U_{(i)}} \right) \\
&= O_p\left\{ \max\left( \frac{n^2}{k^3}, \frac{n^2(k - k(i))^2}{k^4} \right) \right\}.
\end{aligned}
$$

## Footnotes

*Liu and De Brabanter are with Iowa State University, Ames, IA 50011, USA. Corresponding authors: `yuliu@iastate.edu, kbrabant@iastate.edu`

## References

[1]  H.A. David and H.N. Nagaraja. *Order Statistics, Third Edn.* John Wiley & Sons, 2003.