[Reviews · NeurIPS 2018]

Reviewer 1



This paper looks at derivative estimation in random design, where the function is sampled from randomly chosen input points rather than at fixed intervals. While previously proposed estimators have primarily been analyzed in the context of fixed design, this paper presents the natural extension of the algorithm to random design and provide accompanying analysis. A explained by Lemma 1 and the discussion that follows it, a simple estimator in eq (3), which simply computes the local ratio of the difference between output and input variables, has high variance, since the difference between outputs has a noise term with variance 2 \sigma_e^2, and the denominator scales as 1/n, such that the variance of the ratio scales as O(n^2). I'm a bit confused why the authors state that the variance of this method is asymptotically worse for random design? Doesn't the O(n^2) variance of estimate also hold in the fixed design case, such that the high variance is not due to random design but rather simply due to the fact that estimator (3) only computes difference over a single pair of points rather than averaging over many points such that the noise term will affect the estimate greatly? I think the main contribution of the paper is to analyze the algorithm in eq (5) which was proposed in [21] and [22] for fixed design, but to extend the analysis to random design by specifying the weights in eq (6) and then analyzing the asymptotic behavior of the difference of uniform order statistics (Lemma 1), which is then used to show bias and variance bounds on the estimator as in section 2.2, and show how to select the bandwidth k via a data-driven estimates. I was confused by the discussion in 2.6 and did not really understand how the discussion related to the previously described algorithms and analysis. Is it proposing a modified algorithm which uses a different kernel for computing the weights w_ij instead of the one presented previously? Just to clarify, does the correlation function kind of impose a gaussian process like assumption, where points that are nearby have noise terms that are more correlated vs points far away are less correlated? and doesn't that make derivative estimation more difficult since locally you will fit to the noise function as well instead of learn the underlying expected function r? ************************** added after rebuttal ***************************************** Thanks for the clarifications.

Reviewer 2



Paper 1768 This paper considers the univariate regression setting Y_i = r(X_i) + epsilon_i for X_i taking value in [0,1] and independent epsilon_i. The authors study the estimation of the derivative r'(x) of the regression function for x in the interior (0,1). The authors use a weighted finite difference ratio estimator and analyze its behavior in the random design setting where X_i has the uniform density. The main result yields a pointwise rate of convergence for the estimator on the data points X_2, ..., X_{n-1}. Extensions to non-uniform design points are discussed. I find the problem and the results interesting. I think the paper is close to a state where it can be published but it needs to be developed some more before it is ready. I like the results but I think they are a bit limited in scope and leave too many immediate natural questions unanswered. The main result characterizes the bias and variance of the estimated derivative at a data point X_i; it is natural to think about the behavior of the estimator at an arbitrary point x in the interior of of (0,1). I would guess that the estimator still behaves well under the same bounded second derivative condition that the authors imposes on the regression function r. Another natural question is whether the estimated derivative is asymptotically normal. Theory aside, the expression (3) immediately leads one to think about kernel weighing. I hypothesize that using a kernel weighted version of (3) gives an estimator whose behavior is similar to that of one proposed by the authors. Finally, in the case where X is non-uniform but has a density that is bounded away from 0, I would imagine that the exact same method should work with the same, up to a constant, bounds on the bias and variance. It seems excessive in the case of non-uniform X to follow what the authors suggests in section 2.7, which involves estimating the density itself. I understand it is unfair for a reviewer to speak on and on about what the authors could do. I raise these open questions because I think they are natural and not too difficult to address. I could be wrong of course, in which case, I would be happy if the authors could discuss why they are difficult. A second issue I have with the paper, which is not a huge issue, is that the mathematical statements are loose. Some small degree of mathematical rigor is needed for a NIPS statistical methodology paper and this paper does not quite reach that standard. In the statement of theorem 1, it does not make sense to condition on the samples U because that yields the fixed design setting. I think what the authors intend is E_U[ \hat{Y}_i - r'(U_i) | U_i], that is, the expected estimate at the point U_i fixing only the value of the point U_i. I think it is clearer to have an unconditional statement, which holds by just taking an iterated expectation. I list a few other issues. + In expression (4), it is very strange to have the subscript on Y be just i'' and to have the subscript on U be (i)''. I suggest saying something like, by reindexing if necessary, assume that U_1 \leq U_2 \leq ... . + In Section 1.1, it is not technically correct to write m(x) = E(Y) and m(X) = E[Y | X]. The correct expression is m(x_i) = E[Y_i] in the first case and m(x) = E[ Y | X = x] in the second case. + The authors say that we require P(U_(l) = U_(m)) = 0'', but that is already true since the U's are uniformly distributed on [0,1]. I suggest saying something like, we observe that because U_i has a density, P(U_(l) = U(m)) = 0 and thus the estimator is well-defined on U_i for all i = 2, ..., n-1''. + I recommend giving an explicit rate in corollary 1, with k chosen optimally. + Corollary 2 isn't quite right because the the first term is based on an upper bound of the bias. + The authors say in line 202 that they need the existence of a density in order for expression (10) to hold but they in fact need the existence of a density earlier. The distribution of X_1 needs to be absolutely continuous with respect to the Lebesgue measure in order for F(X_1) to be uniformly distributed. If the distribution of X_1 has a component singular to the Lebesgue measure, then the quantile function is not the inverse of the distribution function F and F(X_1) is not uniformly distributed. + I don't know of a result in [1] that describes the optimality of derivative estimators. [1] mostly deals with the estimation of an additive regression function itself. It would be helpful if the authors could provide a reference to the particular theorem in the paper. [1] C. Stone. Additive regression and other nonparametric models. EDIT: I have read the authors' response. I agree that the conditional on U statement as it appears in the paper is actually correct; I had misunderstood it before. I also agree that F needs to only be continuous and not absolutely continuous as I had mistakenly stated before. I apologize for these mistakes. I do not have a reference for using kernel weighing. It just strikes me that the authors' weighing scheme resembles kernel weighing where the kernel is K(u) = 1/u^2 * I(u \leq 1). See an analysis of this kernel in [1]. Bandwidth selection is certainly an issue but so is the selection of "k" in the authors' scheme. I still do not understand why estimating the density f(X) is necessary. It seems that Lemma 1 would hold if X is non-uniform but has a density that is bounded away from 0. It would be great if the authors could comment on this in the paper. I don't think that studying the behavior of \hat{r} is "sufficient for another paper", especially since the authors already use the random design setting. I think not being able to say anything about the behavior of \hat{r} except on the sample points is the most significant weakness of the paper. I have upgraded my assessment of the paper though I still recommend that the authors take some time to further develop the paper before submitting it for publication. But, I don't have very strong objections to accepting the paper as it is. [1] Belkin, M, Rakhlin, A, and Tsybakov, A. (2018) Does data interpolation contradict statistical optimality? Arxiv Preprint: 1806.09471

Reviewer 3



Short summary: In this paper, the authors consider the problem of estimating the derivative of a regression function based on difference quotients in random design - previous methods either assumed fixed design or did not use difference quotients. More specifically, based on the assumption that the $n$ design points are uniformly distributed, an estimator in the form of a weighted sum of difference quotients is proposed and analysed. The authors focus on the order of its bias and variance with respect to $n$ and an additional parameter $k\leq n/2$ of the method to understand its asymptotic behaviour. They also propose a strategy for choosing $k=k(n)$ such that the method's AMISE is small. Furthermore, it is explained that the natural extension to higher order derivatives (in a recursive manner) is not worthwile since it results in a non-consistent estimator. The authors describe how the method can be applied in the situation where the design points are not uniformly distributed (through the integral transform method) - in practice, estimating the derivative then requires the preliminary step of estimating this distribution. Finally, simulation results are provided - in the experiments, the performance of the new estimator is similar to the performance of two previously known techniques. Comments: I enjoyed reading this paper since it is well-written, well-structured and the proofs are relatively easy to follow. The results appear technically sound and original. Moreover, the paper's contribution to the field is discussed in a very honest manner. However, I do have two minor remarks: (i) I would appreciate an intuitive interpretation of the weights from equation (6): At first sight, it is surprising that the weights are growing with $j$ since you expect "better information" the closer you are to $U_{(i)}$. (ii) For asymptotic statements, you typically require that $n,k\rightarrow\infty$ in a certain sense (which is of course reasonable); however, I feel that some clarification of the role of $i$ is warranted: since $k\leq i-1$, you also need $i\rightarrow\infty$. This could be important, for instance, in Corollary 1. Clearly, the main concern about the paper would be its significance: one could argue that the paper does not provide a new method which outperforms previous methods in a general sense and that no novel theoretical tools are introduced. However, I feel that it is worthwile investigating the performance of the natural quantity - difference quotients - in a random design setting and that the analysis is interesting in its own right. Hence, especially in light of the positive aspects mentioned above, I would like to see this paper published.